# Gaze Behavior in Basketball Free Throws Developed in Constant and Variable Practice

**DOI:** 10.3390/ijerph16203875

**Published:** 2019-10-12

**Authors:** Stanisław H. Czyż, Martin Zvonař, Zbigniew Borysiuk, Jiří Nykodým, Piotr Oleśniewicz

**Affiliations:** 1Department of Sport Didactics, University School of Physical Education in Wrocław, 51-612 Wrocław, Poland; 2Physical Activity, Sport, and Recreation focus group, North-West University in Potchefstroom, 2520 Potchefstroom, South Africa; 3Faculty of Sport Studies, Masaryk University in Brno, 625 00 Brno, Czech Republic; zvonar@fsps.muni.cz (M.Z.); nykodym@fsps.muni.cz (J.N.); 4Faculty of Physical Education and Physiotherapy, Opole University of Technology, 45-758 Opole, Poland; z.borysiuk@po.opole.pl; 5Faculty of Physical Education, University School of Physical Education in Wrocław, 51-612 Wrocław, Poland; piotr.olesniewicz@awf.wroc.pl

**Keywords:** skill acquisition, gaze behavior, practice conditions, especial skill

## Abstract

There are a limited number of studies focusing on the mechanisms explaining why variable practice gives an advantage in a novel situation and constant practice in performance in trained conditions. We hypothesized that this may be due to the different gaze behavior that is developed under different conditions. Twenty participants, randomly assigned to two different groups, practiced basketball free throws for three consecutive days, performing 100 throws per day. The constant group (*n* = 10) practiced at a free throw distance (4.57 m) only. The variable practice group (*n* = 10) randomly performed 20 shots per five throw distances (3.35, 3.96, 4.57, 5.18, and 5.79 m) on each day, also accumulating 100 shots per day. We analyzed the total gaze fixation duration, a number of fixations, and the average fixation duration on a basketball rim in a pretest and posttest at the 4.57 m distance. We computed a linear mixed model with test (pretest–posttest), group (constant–variable), and test × group interaction in order to analyze the total fixation duration and number of fixations. The average fixation duration was analyzed with a repeated measure two-way ANOVA, with practice conditions as a between-participants factor and test type as a within-participants factor. We found that the total fixation duration increased significantly in the posttest, regardless of the practice conditions (*p* < 0.001, effect size = 0.504). The number of fixations also increased significantly in the posttest (*p* = 0.037, effect size = 0.246). The average fixation duration increased in both groups; however, insignificantly. We also did not find any significant differences between groups. Our results suggest that variable and constant practice conditions may lead to the development of similar gaze behavior.

## 1. Introduction

The role and benefits of different practice conditions have been studied for decades. Practice conditions can have an impact on the capability of performing a skill during games and competitions. One of the practice characteristics is the variability of practice, which refers to the variety of skills and context characteristics that can be manipulated during practice [1]. Variability of practice may and does influence motor learning. This influence is best understood from theoretical viewpoints. For example, schema theory [2] has been used to explain the benefits of variable practice, whereas Thorndike’s theory of identical elements [3] has been used to explain the benefits of constant practice. Moreover, the benefits of practice conditions are well-supported by experimental evidence (e.g., for variable practice, see [4,5]; for constant practice, see [6,7]).

There is currently a consensus that variable practice, involving the practice of several variations of a skill, benefits learning differently than practice in constant conditions, i.e., practice that involves only one variation of a skill. It is assumed that, favoring generalization, variable practice results in better retention and transfer [5] and in a more accurate and stable performance [8,9]. On the other hand, practice in a constant condition makes the recall schema more refined [6,8]. As a result, massively trained variation of a skill, called especial skill [6], produces an advantage in performance compared to the same variation of skill practiced in variable conditions (assuming that practice in variable and constant conditions has a similar capacity) [5]. It also produces an advantage in performance over other actions in the class of actions executed by the same Generalized Motor Program (GMP) [2]. In practice, it means that constant practice makes the trained variation of a skill the best performed within the whole class. This effect dissipates if some variability in practice is introduced [10].

The question raised by Keetch et al. [6] was why constant practice gives an advantage to trained variation of the skill over non-trained variation. Several studies have attempted to address this issue, e.g., [11,12,13]. It is important to emphasize that the findings reported by Keetch et al. [6] are in contrast to Schmidt’s schema theory [14] and the evidence provided by early studies on the variability of practice [4]. Schmidt’s schema theory [2] assumes that practice, even constant practice, should contribute to the development of the whole schemata. Indeed it does [8], but it contributes to the development of a trained variation of skill the most [6,8,12,13]. The latter finding contradicts the schema theory, which assumes that it should contribute to the development of the whole schemata equally.

To the best of our knowledge, there has been no behavioral study trying to determine why the performance of a variation of the skill trained in constant conditions is better than the performance of the same variation of the skill trained in variable practice (assuming that the amount of training for that skill is the same in both conditions). There are studies focusing on neurophysiological aspects of the variability of practice, e.g., [15], or on the influence of random and blocked practice on gaze behavior [16,17]. However, the first study did not explain the potential influence of variability of practice from a behavioral perspective, instead focusing on how the practice is correlated with neurobiological substrates. The latter studies looked at contextual interferences which may involve the practice of skills governed by different GMPs rather than at practice involving variations of one skill executed by the same GMP (called variable practice conditions).

At this point, we do not know whether different practice conditions affect gaze behavior and performance differently. We assumed (hypothetically) that practicing only one variation of a skill can develop performer “expertise” on the trained variation of a skill (e.g., especial skill in basketball free throws [6]) compared to non-trained variations. This is why especial skill may be outperformed as compared to skills from the same class of actions. A similar situation is noticed when comparing variation of a skill trained in constant and variable practice—the former is always better [18,19,20].

On the other hand, there is strong evidence that expert performance is associated with a different gaze behavior compared to novice performance [21,22]. Usually, experts look at the target area for a longer period of time than novices [22] and as the practice progresses [23,24]. Therefore, the primary objective of this study was to determine how two types of practice (constant and variable) affect gaze behavior. Given that our study is an exploratory study in nature and there is no existing research, it is difficult to predict the direction and magnitude of differences between groups practicing in different conditions. However, for the purpose of this study, we advanced the hypothesis that practice in a constant condition will result in different gaze behavior compared to practice in variable conditions. We decided to analyze fixations, since they are generally deemed the best determinants of success in single fixed-target aiming tasks [24], as well as the most popular variables, analyzed in sport [25]. Therefore, fixations on a basket’s rim (regardless of whether it is the front, back, or middle) [24] seemed the most appropriate measure of gaze behavior differences in our study. We decided to look at the total fixation duration per shot, average fixation duration, and numbers of fixations per shot.

We applied the procedure of Breslin et al. [8]. In their study, Breslin and colleagues recreated the effect of especial skills after 300 trials of free throws in constant practice conditions [8]. Especial skill is a trained variation of a skill that is significantly outperformed compared to other, non-trained variations. Both trained and non-trained variations are executed and governed by the same GMP. We decided to follow this procedure as we believed that if there were any advantages in regards to the gaze behavior followed by practice in constant conditions, they would be the most conspicuous in especial skill or after practice which leads to the emergence of especial skill. If constant practice gives any advantages to the gaze behavior of a trained variation of skills compared to the same variation developed in variable practice conditions, they should be noticeable after 300 trials. Therefore, the secondary objective was to develop especial skill in constant practice conditions.

The results of our study may contribute to general knowledge on how motor skills that develop in variable and constant practice are governed, executed, and controlled. Moreover, if there are differences between gaze patterns in constant and variable practice conditions, one could focus on perceptual training supporting the development of beneficial gaze behavior. Manipulation of the practice conditions could lead to the development of specific behavior.

## 2. Materials and Methods

Permission for the subjects to participate in the experiment described in this manuscript was specifically approved and granted by the Health Ethics Research Committee at North-West University, South Africa (ref. number: NWU-00180-15-A1). All participants took part in the study on a voluntary basis and could discontinue their participation at any time, without any consequences. The written participants’ informed consent was obtained by a trained, independent, and bias-free person.

### 2.1. Participants

We assumed that if the differences between the constant and variable conditions practice have to be conspicuous, participants should reach a difference in shooting accuracy in constant conditions similar to that reported by Breslin and colleagues [8]. We used means and standard deviations for pre- and posttest average shooting accuracy, as presented in Table 1 [8], where the confidence interval was set to 95% and the power to 0.8. The sample size was calculated using Open Epi software, version 2(Open Source Epidemiologic Statistics for Public Health; www.OpenEpi.com). Our sample size per group (10 participants per group) was higher than estimated (seven participants per group) to prevent problems with potential dropout.

Twenty young, healthy, right-handed males participated in the study. In order to minimize the influence of previous ball-shooting experience on the study outcome, only participants with no previous experience in basketball were included, in order to minimize a potential transfer from previous basketball experiences to the experimental settings. We assumed that participants who had previous training in basketball could have acquired specific shooting skills and looking patterns. We also considered that although basketball is not very popular in South Africa, there are no physical education classes provided in schools during which participants could have played basketball and the basketball infrastructure is very poor, but two other “basket” games are very common, i.e., korfball or netball. These two games are specifically popular among females, so we decided to recruit males only. Additionally, only participants with no previous experience in korfball or netball were included. We defined “previous experience” as “having been involved in organized training supervised by a trainer, instructor or a coach at least two times a week for at least three consecutive months”.

Participants were randomly allocated into two groups: 10 participants to the constant group (CG, M age = 21.80; SD = 0.79) and 10 participants to the variable group (VG, M age = 22.70; SD = 1.42). Participants from both groups were actively involved in other sports: on average, participants from the CG had played rugby (7 participants) and cricket (3 participants) for 151.10 months (SD = 31.59), whereas participants from the VG had played rugby (4 participants); cricket (2 participants); and tennis, field hockey, squash, and javelin throw (one participant per sport) for an average of 158.90 months (SD = 187.20).

### 2.2. Procedure

We adopted the experimental procedure presented by Breslin et al. [8].

On the first day of testing, the participants were required to complete a short questionnaire consisting of personal information questions regarding each player’s name, age, exercise habits, injury occurrences, and years of participation in other sports. They were also familiarized with the eye tracking device that was used during testing procedures prior to any testing and training.

The pre and posttest were performed on day 1 and day 3, respectively. Both tests consisted of shooting proficiency measurements: Twenty shots per five different distances (3.35, 3.96, 4.57, 5.18, and 5.79 m) were taken. The first 15 shots per each distance were taken in a quasi-random order such that no more than two shots were performed from the same distance in consecutive trials and a sequential presentation was avoided. The last five shots were performed with the participants wearing eye-tracking glasses. In order to optimize the testing procedure, i.e., minimize the time needed for adjusting the eye tracker, calibration of devices, mounting of glasses, etc., the last five shots per distance were performed in a blocked order.

During the acquisition phase, which started immediately following the pretest on day 1, on day 2, and immediately preceding the posttest on day 3, participants were required to practice basketball shooting consisting of 100 throws per day. Participants from the CG practiced for three consecutive days, in constant conditions. They performed free throws from a free throw distance (4.57 m) only, accumulating a total of 300 shots in 3 days. Participants from the VG practiced for three consecutive days, in variable conditions, i.e., on each day, they randomly performed 20 shots per distance (100 throws a day in total) from five throw distances (3.35, 3.96, 4.57, 5.18, and 5.79 m), accumulating a total of 300 shots in 3 days.

A 5 s rest interval was provided between each shot during testing and training sessions. Prior to testing and training, participants were engaged in a general 5–10 min warm-up consisting of low-intensity aerobic activities, followed by static and dynamic stretches.

### 2.3. Apparatus

Participants were tested and trained using an official basketball (size 7, Spalding) on a standard basketball court. The basketball court, the board, the rim, and the free throw distance were set according to the International Basketball Federation (FIBA–Fédération Internationale de Basketball Amateur) regulations, i.e., the rim was mounted on a standard basketball board at the height of 3.05 m from the floor and the 4.57 m line on the basketball court was the foul line (free throw line). The shooting distances, except for the free throw distance (4.57 m), were marked along a straight line radiating from the backboard toward the center of the court with strips of masking tape 2 cm wide and 5 cm long. Participants were encouraged to perform shots with their preferred limb and without any pre-shot movements (e.g., dribbling or bouncing). They were also asked to perform shots with their feet close to the marking tape, maintaining contact with the floor at all times.

Gaze behavior was registered using the mobile eye-tracking system Dikabilis (Ergoneers GmbH, Bavaria, Germany). The system consisted of eye camera tracking binocular glasses (pupil tracking accuracy: 0.05° visual angle; 60 Hz per eye; resolution up to 648 × 488 pixels) and a head-mounted scene-camera (glance direction accuracy: 0.1–0.3° visual angle; resolution: 1920 × 1080 pixels; full HD; aperture angle: 40–90°). The recordings were analyzed using Dikablis Professional software (D-Lab. 3.01.7550).

Orthogonal to the shooting plane, a widescreen digital high-definition camera (Logitech HD; resolution: 1280 × 720 pixels) was mounted on the tripod at the height of the shoulders, head, and raised arms of the performing participant, in order to determine the moment of ball release [26,27].

### 2.4. Data Analysis

The shooting proficiency was calculated using a two-point scoring system [6,13,28,29], i.e., 1 point was awarded for a successful shot and 0 for missing a basket.

Gaze behavior was analyzed frame by frame, starting from the first moment the illumination of the flashlight was recorded by a head-mounted scene-camera and the gaze left the flashlight and was directed at any other locations [30], until the ball was released [27]. We analyzed gaze fixations held on one location (basketball rim) within 3° of a visual angle (foveal vision) or less for a minimum of 100 ms [24]. Given the frequency of our eye-tracking system (60 Hz), we considered the only fixation on a target for at least six frames (16.667 ms/per frame × 6 frames = 100.02 ms; the exact frame duration is 1 frame ≈ 0.0166 s). We analyzed the total fixation duration per shot, average fixation duration, and number of fixations per shot.

The total fixation duration per shot and number of fixations per shot were analyzed with a linear mixed model with a random intercept (participants treated as a random effect), test (pretest–posttest), group (CG–VG), and test × group interaction (treated as fixed effects). The variance/covariance of the 10 shots per participant (5 pre-, 5 posttests) was taken into account within the unstructured covariance matrix and in the calculation of the variance that was used in the effect size calculations. It could be interpreted similarly to Cohen’s *d* values. We applied Bonferroni adjustments for multiple comparisons.

The differences between the average fixation duration were analyzed with a repeated measure two-way ANOVA, with practice conditions (two levels: CG and VG) as a between-participants factor and test type (two levels: pre- and posttest) as a within-participants factor. Partial eta squared was used as the effect size measure.

## 3. Results

In order to meet the objectives of the study, we analyzed our data from two different perspectives. One focused on detecting especial skills, and the other on the gaze behavior. The gaze behavior was analyzed for 19 participants: one participant from the CG was removed as the pretest results were not available due to eye-tracker software problems (no fixation points).

### 3.1. Especial Skills

We used Keetch et al.’s [6] methodology (Exp. 1) to detect especial skills in our participants. Average percentage accuracy scores (based on a two-point scoring system) were calculated for the 20 trials conducted across each of the five distances (Table 1). These scores were used to compute regression lines for all of the shooting distances but the free throw distance, i.e., 4.57 m for each participant (regression equations and R^2^ values provided in Figure 1). We calculated the expected values using personal regression equations. Finally, we compared expected and real means using two-tailed paired *t*-tests.

As presented in Figure 1, especial skills did not emerge in the posttest in any of the training conditions. In the CG, the expected values were higher than the real ones (t(9) = 0.23, *p* = 0.82; real M = 27.00, SD = 10.85; expected M = 28.13, SD = 8.92). Similar results were noticed in the VG (t(9) = 0.33, *p* = 0.75; real M = 20.00, SD = 11.30; expected M = 20.87, SD = 4.86). In order to ensure that especial skills were not present before practice, we also compared real and expected means in both groups for the pretest. In the VG, the real value was slightly higher than the expected one, although the difference did not reach a significant level (t(9) < 0.001, *p* = 0.99; real M = 16.00, SD = 10.74; expected M = 15.99, SD = 5.88). Additionally, there was no significant difference between real and expected means in the CG ((t(9) = 1.98, *p* = 0.07; real M = 18.50, SD = 12.48; expected M = 25.99, SD = 8.74).

### 3.2. Gaze Behavior

We computed linear models using the total fixation duration per shot and number of fixations per shot as dependent variables in the first and second model, respectively. An analysis of gaze behavior revealed that the total fixation duration significantly increased in the posttest compared to the pretest (see Table 2). We also found that the interaction effect, i.e., interaction between practice conditions and pre- and posttest, was significant (*p* = 0.008). Further pairwise comparison showed that the total fixation duration significantly increased in CG in the posttest compared to the pretest (Table 3 for references), and the effect size was large (*ES* = 0.80). CG and VG groups did not differ one from another in either the pre- or posttest.

We also observed significant differences between pre- and posttest results while analyzing the number of fixations (Table 4). However, the effect size was small (*ES* = 0.27). We did not find significant differences between groups or a significant interaction effect. In the pairwise comparison (Table 5), the only significant difference was found between pre- and posttest results in CG, but the effect size was small (*ES* = 0.44). In the posttest, the number of fixations per shot significantly increased.

We did not find any significant (*p* > 0.05) differences in the analysis of average gaze duration. A two-way repeated measure ANOVA (F(1, 17) = 1.42, *p* = 0.25) yielded insignificant main effects (CG vs. VG; pretest vs. posttest) and interaction effects (group × test) (Table 6). The average gaze duration increased insignificantly in posttests in both groups compared to the pretest (Table 7). The changes between the pre- and posttest in both groups are depicted in Figure 2 and the individual average gaze durations in Figure 3.

### 3.3. General Assumptions

We tested whether the shot duration (total time of the shots recorded) was independent of practice conditions (variable vs. constant), test type (pretest vs. posttest), or interaction effects. We computed the repeated measure analysis of variance, 2 × 2 ANOVA with practice conditions (two levels: CG and VG), as a between-participants factor and test type (two levels: pre- and posttest) as a within-participants factor. The overall shot duration was independent of the practice conditions (F(1.17) = 0.97; *p* = 0.34, η^2^ = 0.05); test type (F(1.17) = 3.81; *p* = 0.06, η^2^ = 0.18), as well as the interaction effect (F(1.17) = 1.91; *p* = 0.18, η^2^ = 0.10).

Wearing eye-tracking glasses did not affect the shot proficiency. The shot proficiency in the pretest was 15.00% (SD = 22.36) while wearing the eye tracker and 18.00% (SD = 10.39) while not wearing the eye tracker. The difference was not significant (t(19) = 0.66, *p* = 0.51). Furthermore, there was no significant difference in posttest results between the shot proficiency while wearing (23.00%, SD = 19.76) and not wearing the eye tracker (24.67%, SD = 12.99) (t(19) = 0.37, *p* = 0.74). Both these results suggest that wearing the eye tracker had no significant effect on the shooting proficiency.

We also tested whether an improvement in shot efficiency at the 4.57 m distance (= shot efficiency in the posttest–shot efficiency in the pretest) correlated with a prolonged total gaze duration (=total gaze duration in posttest–total gaze duration in pretest). It did not correlate significantly (*p* > 0.05) in CG (r = −0.22) or VG (r = −0.35).

## 4. Discussion

The primary objective of this study was to determine how two types of practice (constant and variable) affect gaze behavior, whereas the secondary objective was to develop especial skill in constant practice conditions and replicate the results obtained by Breslin et al. [8].

Our main finding, with respect to the primary objective, was that the total fixation duration at the target was significantly longer in the posttest than in the pretest. We did not find any differences between the CG and VG group. We also found that there was a significant interaction effect (test × group) when the total fixation duration was used as a dependent variable. Additionally, we found that the number of fixations increased in the posttest, although the effect was small (ES = 0.246). The average fixation duration increased insignificantly in posttests in both groups. This finding is in line with results presented in previous studies on gaze behavior in basketball free throws: The elite participants look at the target longer [31] compared to the near-elite. In general, novices look at the target area for a shorter duration (for a review, see [21]).

Usually, looking at a target area for a longer period of time improves the shot efficiency at the free throw distance [32]. However, we did not find a correlation between the total fixation duration and shot efficiency at the free throw distance in either of the groups. Hence, we had a more advanced gaze behavior (longer total fixation duration), with no correlation to the higher shot efficiency. An explanation for this may be provided by a few theories. In general, at the very early stage of learning, changes occur in the cognitive system prior to changes in performance (motor system) [33]. Therefore, it may be that cognitive changes, such as longer fixation duration, occur earlier than shot proficiency improvement.

Given that looking longer is linked to a higher shot proficiency, we could assume that similar gaze behavior, i.e., looking at the target area for longer, was present at the non-trained distances in CG. This is because we did not detect an especial skill effect in the constant practice group. This means that shots at the free throw distance were not outperformed compared to the shot efficiency at other distances. In order to maintain the same shot efficiency, participants had to demonstrate similar gaze behavior at all distances: non-trained and trained. Our finding is in line with previous studies, especially on intra-task transfer, which was observed in elite performers [34,35].

Given that looking at the target area for longer is associated with fine-tuning and programming [36], we could assume that in trained and non-trained variations of a skill (in CG), these processes took the same amount of time. Since it is quite possible that parametrization (also fine-tuning) is the dominant process constituting the especial skill effect [7,29], it would be very interesting to see if the gaze behavior changes while the learning progresses and whether the fixations are substantially longer in especial skill as opposed to other non-trained variation of that skill. It could be asked if there would be differences in gaze behavior of a trained variation of the skill developed in constant and variable practice conditions at later stages of learning [37]. This question could be addressed in future studies.

Our secondary objective was to develop especial skill in constant practice conditions. Especial skill did not emerge in our participants, i.e., we did not confirm that constant practice conditions give an advantage to the trained variation of skills compared to the non-trained and trained in variable conditions. Although our participants received the same amount of practice as in the Breslin et al. [8] study, there was no sign of superiority in performance at the 4.57 m distance in the CG. The reason why we did not recreate especial skill in CG could have been due to the manner in which the testing was conducted. Our participants performed eye-tracking testing in a blocked manner, i.e., they performed five shots at the 4.57 m [5] distance, whereas Breslin et al.’s [8] participants performed it in a random manner. The practice test schedule affects both transfer and retention in motor learning [38]. There might be many other reasons why especial skill did not emerge in our participants [39] and these did not form the focus of this study.

The limitations of this study has to be acknowledged. First of all, we did not notice an especial skills effect. As a result, it is more difficult to explain our findings on a theoretical ground. We initially assumed that the emergence of especial skill would enhance the advantages of the practice in constant conditions, thus resulting in more obvious and conspicuous differences between the same variation of a skill practiced in constant and variable conditions. If we had detected especial skill, and there were no differences between gaze behavior in CG and VG, we could have claimed that it is consistent with Schmidt’s schema theory [2] since it assumes that practice develops the whole schema, regardless of whether it is practice in constant or variable conditions. On the other hand, if there had been differences, we could have concluded that it is in line with Thorndike’s transfer theory [3] and with the intra-task transfer specifically [34,35]. Given that we did not notice especial skill, the reasoning is more problematic. Second, we analyzed gaze behavior utilizing a task with a fixed target. The result may vary in a task with a target in motion or with an abstract target.

One could use the aforementioned limitations as objectives in follow-up studies. In our opinion, the results of the next studies addressing these concerns may clarify the role of gaze behavior in performance/training under different conditions.

## 5. Conclusions

We found that practice in constant and variable conditions develops a similar gaze behavior pattern. The strength of this study is that, to the best of our knowledge, it is the first study focusing on gaze behavior developed in constant and variable practice.

## Figures and Tables

**Figure 1 ijerph-16-03875-f001:**
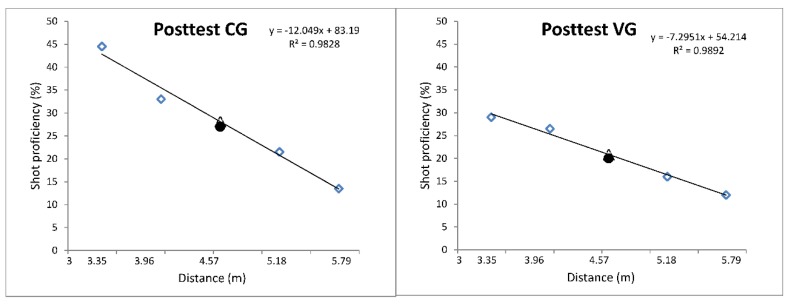
Posttest average shot proficiency across the 3.35, 3.96, 4.57, 5.18, and 5.79 m distances from the basket. The non-filled triangles illustrate the expected shot proficiency values at the 4.57 m distance, based on linear regression. Filled black circles represent real values (shot proficiency) at a distance of 4.57 m from the basket. The non-filled diamonds represent mean shot efficiencies at all distances but 4.57 m.

**Figure 2 ijerph-16-03875-f002:**
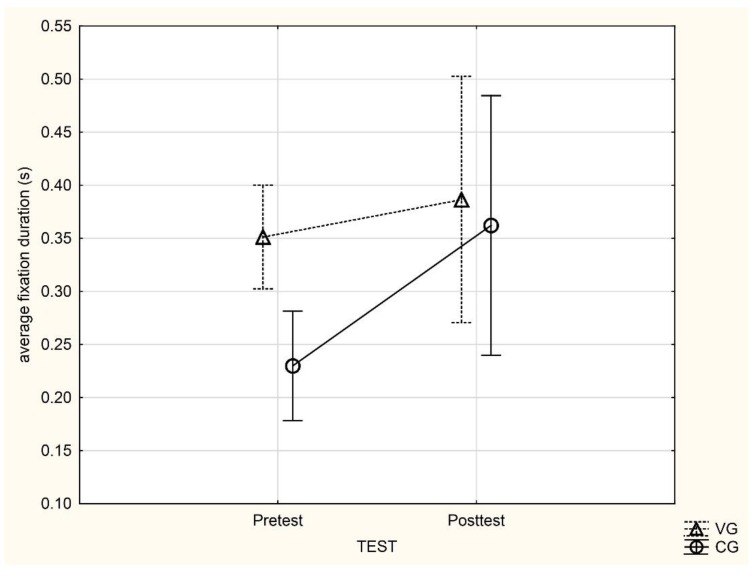
The changes in average gaze duration between pre- and posttest in a variable group (VG) and constant group (CG) practice conditions group.

**Figure 3 ijerph-16-03875-f003:**
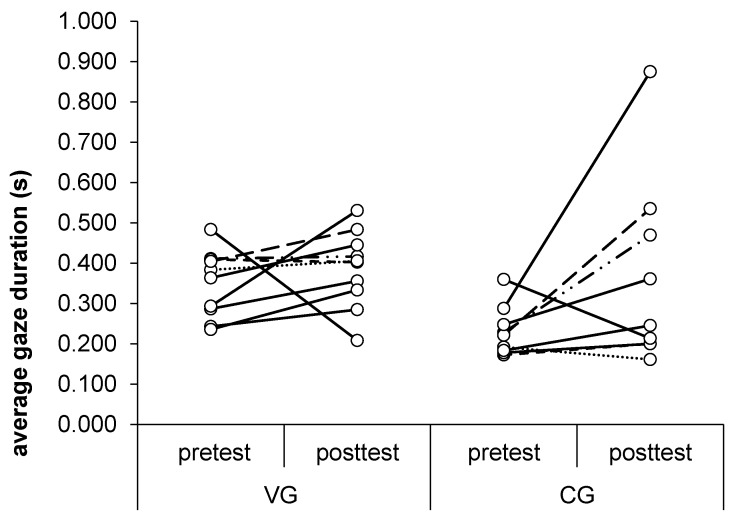
The average gaze duration in both groups in the pre- and posttest for each participant. Lines connect corresponding values for each participant.

**Table 1 ijerph-16-03875-t001:** Average percent accuracy scores for each distance in the pretest and posttest in variable group (VG) and constant group(CG) practice conditions.

Distance (m)		Pretest					Posttest				
		3.35	3.96	4.57	5.18	5.79	3.35	3.96	4.57	5.18	5.79
VG	Mean	24	20	16	11	9	29	26.5	20	16	12
	SD	7.38	11.30	10.75	5.68	8.76	10.75	9.73	11.30	6.15	5.37
CG	Mean	39	30.5	18.5	20.5	14	44.5	33	27	21.5	13.5
	SD	17.29	11.89	12.48	7.62	10.49	15.54	16.02	10.85	10.55	8.18

**Table 2 ijerph-16-03875-t002:** Results of the linear mixed model of gaze behavior. Dependent variable: total gaze duration. *F*-values, *p*-values, degrees of freedom (*df*), and effect sizes (*ES*) for main effects (group: CG vs. VG; test: pretest vs. posttest) and interaction (test × group).

Source	Numerator *df*	Denominator *df*	*F*	*p*	*ES*
Intercept	1	17	61.56	0.000	
Group	1	17	0.45	0.51	0.21
Test	1	169	20.00	0.000	0.50
Test × Group	1	169	7.125	0.008	

**Table 3 ijerph-16-03875-t003:** Total gaze duration (seconds) per shot (mean values), *F*-values, degrees of freedom *(df*), *p*-values, and size effects (*ES*) for pairwise comparisons of CG and VG in pre- and posttests.

	Pretest	Posttest	*F*	*df*	*p*	*ES*
CG	0.24	0.52	24.23	17	0.000	0.80
VG	0.42	0.49	1.72	169	0.19	0.20
*F*	2.39	0.07				
*df*	21.72	21.72				
*p*	0.14	0.79				
*ES*	0.51	0.09				

**Table 4 ijerph-16-03875-t004:** Results of the linear mixed model of gaze behavior. Dependent variable: number of fixations per shot. *F*-values, *p*-values, degrees of freedom (*df*), and effect sizes (*ES*) for main effects (group: CG vs. VG; test: pretest vs. posttest) and interaction (test × group).

Source	Numerator *df*	Denominator *df*	*F*	*p*	*ES*
Intercept	1	17	94.99	0.000	
Group	1	17	0.01	0.91	0.03
Test	1	169	4.42	0.04	0.27
Test × Group	1	169	2.89	0.09	

**Table 5 ijerph-16-03875-t005:** Number of fixations per shot (mean), *F*-values, degrees of freedom *(df*), *p*-values, and effect size (ES) for pairwise comparisons of CG and VG in pre- and posttests.

	Pretest	Posttest	*F*	*df*	*p*	*ES*
CG	1.02	1.40	6.86	169	0.01	0.44
VG	1.22	1.26	0.09	169	0.77	0.05
*F*	0.53	0.27				
*df*	22.67	22.67				
*p*	0.47	0.61				
*ES*	0.23	0.16				

**Table 6 ijerph-16-03875-t006:** Results of two-way repeated measure ANOVA with practice conditions (two levels: CG and VG) as a between-participants factor and test type (two levels: pre and posttest) as a within-participants factor. Provided: *F*-values, *p*-values, and partial eta-squared (*η^2^*) as a measure of the effect size.

Effect	*F*	*p*	Partial *η^2^*	Observed Power (Alpha = 0.05)
Intercept	210.28	0.000	0.92	1.00
Group	2.52	0.13	0.13	0.32
Test	4.24	0.055	0.20	0.49
Test × Group	1.42	0.25	0.08	0.20

**Table 7 ijerph-16-03875-t007:** Average fixation duration (seconds), mean values, standard deviations (*SD*), and confidence intervals.

Group	Test	N	Mean	SD	−95.00%	95.00%
VG	Pretest	10	0.35	0.02	0.30	0.40
VG	Posttest	10	0.39	0.05	0.27	0.50
CG	Pretest	9	0.23	0.02	0.18	0.28
CG	Posttest	9	0.36	0.06	0.24	0.48

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
