# Peer review of "Gaze Behavior in Basketball Free Throws Developed in Constant and Variable Practice"

_ijerph, 2019, doi:10.3390/ijerph16203875_

Round 1
Reviewer 1 Report
Gaze behavior in basketball free throws developed in 2 constant and variable practice
The interest of the topic of this paper is the novelty of analyzing the gaze behavior in basketball free trows in constant and variable experimental conditions. The topic is interesting but its practical applicability ist not very clear in the introduction.
Paper strengths:
The novelty of the subject dealt with, the procedure ist rigourous, the methods are expleined in detail, the statistical analysis is appropiate and the reviewed literature updated.
Paper weaknesses:
The introduction should place grater emphasis on the importance and applicability of the objective to analyse. In the discusión does not make clear the relationshipof the Scheme theory with the results obtained.
There misprints in the bibliography (some initials of the autors’ names in lower case: 6, 8, and the reference 31 is not complet). There are also misprints in the introduction, lines 43, 103, 104)
Author Response
The interest of the topic of this paper is the novelty of analyzing the gaze behavior in basketball free trows in constant and variable experimental conditions. The topic is interesting but its practical applicability ist not very clear in the introduction.
ANSWER: Thank you for your valuable comments. We addressed them below and in the main file (highlighted). We are convinced your comments helped us to improve our manuscript.
Our study could be classified as a basic research. It means that we focused on mechanisms differentiating benefits in two practice conditions: variable and constant. We agree that pragmatic approach is very important, although, we have to know why constant and variable practice conditions benefit performance differently in the first place. We tested a hypothesis that it may be due to different gaze behavior pattern developed in these conditions. We found there is no significant difference. Practically, it means there is no specific gaze pattern that a trainer, coach, skill instructor or a physiotherapist should look at or help to develop in order to benefit performance of a learner.
We elaborated more about possible application in the introduction. (lines 110-114).
Paper strengths:
The novelty of the subject dealt with, the procedure ist rigourous, the methods are expleined in detail, the statistical analysis is appropiate and the reviewed literature updated.
ANSWER: thank you.
Paper weaknesses:
The introduction should place grater emphasis on the importance and applicability of the objective to analyse. In the discusión does not make clear the relationshipof the Scheme theory with the results obtained.
ANSWER: we elaborated more on practical implication in the introduction as suggested (lines 110-114). We clarified the relationship to the schema theory in lines 377+378
There misprints in the bibliography (some initials of the autors’ names in lower case: 6, 8, and the reference 31 is not complet). There are also misprints in the introduction, lines 43, 103, 104)
ANSWER: corrected. Reference 31 – re-written according to IJERPH requirements.
Reviewer 2 Report
General comments
The article “Gaze behavior in basketball free throws developed in constant and variable practice” aimed to compare data within and between groups after an intervention program based on either constant or variable practice of basketball free throws. In general, results indicated that both groups changed similarly the gaze behavior after the intervention program. The study is relevant and provides interesting and innovative insights into the motor learning area of knowledge. However, I believe that the experimental design is not the most adequate for this study. Besides, the dependent variables could be different, and the current ones are weakly justified. In summary, I recommend major revisions, mainly focused on providing clear justifications for some methodological issues, which are pointed out below.
Specific comments
Abstract
Include statistical values for all results presented.
Include the statistical procedures used to compare the data within and between the groups.
Introduction
In general, it is clear and provides us with the most relevant information regarding the area of knowledge. However, it is sometimes too long, so a reduction is desirable. Besides, some minor issues must be addressed.
“We advanced the hypothesis that practice in constant condition will result in different gaze behavior as compared to practice in variable conditions” This is not a valid hypothesis for a scientific study. You should point out what is expected in your dependent variables between and within groups. Just expecting differences are not enough, but the meaning of the difference (if the values are expected to be higher/lower in the pre or post measures or higher/lower on a specific group).
“We decided to analyze fixations since they are generally deemed the best determinants of success in aiming tasks as well as the most popular variables analyzed in sport”. I don’t believe this reason is strong enough. Saccades, quiet eye and, most importantly, the areas of interest, are also very common measures that were not taken into account by you. Specifically, why not include the analysis of the AOI?
Lines 103-103 are probably wrong.
Methods
There is no information about the sample size estimation. This is mandatory information for this type of article.
A control/placebo group is missing in the experimental design. Without this group, the absence of differences between groups (reported later in the article) cannot be suitable. What do you think about including this group?
What is the criterion for choosing these specific distances for the trials?
Why a retention measure was not included?
You included effect sizes in tables 2 and 3, but there is no information about it in this section.
Results
Please, standardize the number of digits after the comma in all tables (I suggest 3).
Without the information about what effect size was calculated, it is impossible to check the effect of the intervention only based on the available numbers.
In general, the order of the tables and the results is not the best. You should point out in the methods section all comparisons made, and at the results section, you should follow the same order, presenting just one table/figure for each analysis. There is a massive number of tables, which, in my opinion, reduces the comprehension of the article.
If the investigation of the correlation between total fixation duration and shot efficiency is an aim of the study, it should be presented much earlier (in the introduction). Considering the current version, no rationale supports this analysis (and no hypothesis is established).
Discussion
I strongly disagree that “practice in both conditions helped to acquire an “expert”-like gaze”. In my opinion, you don’t have enough data to support this affirmation.
Since the choice to make a blocked assessment trial was methodologically justified, you should not use it as a possible explanation for the results (lines 348-352). If this implies problems for you, you should’ve done it differently since the beginning. So, your methodological choices are playing against your results.
Conclusion
Concise, what is desirable in my opinion. However, I am not sure that, without a placebo group, this conclusion is, in fact, withdrawable from the results.
References
The issue is missing in many references.
Author Response
General comments
The article “Gaze behavior in basketball free throws developed in constant and variable practice” aimed to compare data within and between groups after an intervention program based on either constant or variable practice of basketball free throws. In general, results indicated that both groups changed similarly the gaze behavior after the intervention program. The study is relevant and provides interesting and innovative insights into the motor learning area of knowledge. However, I believe that the experimental design is not the most adequate for this study. Besides, the dependent variables could be different, and the current ones are weakly justified. In summary, I recommend major revisions, mainly focused on providing clear justifications for some methodological issues, which are pointed out below.
ANSWER: thank you for your valuable comments. We addressed them below and in the manuscript (highlighted). We are convinced that although we do not agree with some of your comments, re-thinking our design and rationalization significantly improved our manuscript. Thank you!
Specific comments
Abstract
Include statistical values for all results presented.
Include the statistical procedures used to compare the data within and between the groups.
ANSWER: Added as suggested.
Introduction
In general, it is clear and provides us with the most relevant information regarding the area of knowledge. However, it is sometimes too long, so a reduction is desirable. Besides, some minor issues must be addressed.
“We advanced the hypothesis that practice in constant condition will result in different gaze behavior as compared to practice in variable conditions” This is not a valid hypothesis for a scientific study. You should point out what is expected in your dependent variables between and within groups. Just expecting differences are not enough, but the meaning of the difference (if the values are expected to be higher/lower in the pre or post measures or higher/lower on a specific group).
ANSWER: We agree that we may expect a longer fixation duration as the practice progress. It is quite obvious given the existing evidence. However, this is not what we wanted to test. We wanted to test (confirm or falsify) a hypothesis whether different practice conditions would have resulted in different gaze behavior. As I. Jones (2015, Research methods for sport studies, Routledge, 3rd ed.) points out “a hypothesis is essentially a predicted result, based on existing knowledge that can be tested through the collection of data” (p. 21). We should emphasize that our study is exploratory in nature. It means we have no clue of what could be expected. And there is no extant research that can give us a clue about what difference between groups, its magnitude (i.e. effect size and statistical significance) and direction (which group will have more/less fixation or fixation duration) could be anticipated. However, we tried to address your comments. We added, that our study is exploratory in its nature and we advanced the hypothesis only for the purpose of this study (see lines 89-91).
“We decided to analyze fixations since they are generally deemed the best determinants of success in aiming tasks as well as the most popular variables analyzed in sport”. I don’t believe this reason is strong enough. Saccades, quiet eye and, most importantly, the areas of interest, are also very common measures that were not taken into account by you. Specifically, why not include the analysis of the AOI?
ANSWER: We are not sure if we understand this remark correctly. An area of interest (AOI) is a pre-defined area in the environment, which is of interest to researchers. As Kredel et al. (2017) stated “The resulting videos are typically manually analyzed frame-by-frame. The current gaze point is generally allocated to a pre-defined area of interest (AOI; e.g., the opponent's upper body) if either the foveated region overlaps with this area or the gaze is closer to this area than to all other pre-defined AOIs(…). From these allocations and—if analyzed—their dynamics over consecutive frames, (object-related) fixations and eye movements can be determined, in turn, allowing for the derivation of further aggregated gaze variables (fixation duration, number of fixations, saccade-related measures, overall viewing time, dynamics of fixations on AOIs, etc.).”
In other words, AOI analysis is an analysis in which we specify one or more AOI’s and analyze how many times, how long for, etc. a participant fixated on it. In our case, we originally specified the AOI (only one) as following “We analyzed gaze fixations held on one location (basketball rim) within 3° of visual angle (foveal vision) or less” (lines: 202 - 204). Hence, we had one AOI: basketball rim. However, we added what specific area of interest we were keen on in the abstract (line 25) and introduction (line 94 -95) in order to clarify what was it from the very beginning.
We do not agree that the rationale for fixation analysis is not “strong enough”. Kredel et al. (2017) “On the one hand, one finds measures that had been calculated in more than 80% of the studies (with no considerable trends over time). All of these variables refer to fixations, namely fixation duration (FD), number of fixations (NF) and viewing time (VT, additively derived from the FD measure).”
We agree that saccades, smooth pursuit, vergence, pupils size, and many other analyses may be also interesting. However, our participants performed a basketball free throw, which is a single fixed target aiming task (added line 94), and a discrete and closed skill at the same time. The environment in which the task is performed is static. There are no moving objects. And no previous studies indicate that e.g. saccades affect shooting proficiency in basketball free throw. Given existing literature on free throws in basketball, we may assume that fixations are the most important determinants of success in basketball free-throw shooting (see: e.g. Vickers, 2007, Perception, cognition, and decision training; Causer, et al., 2011, Quiet Eye Training in a Visuomotor Control Task; Harle & Vickers, 2001,Training Quiet Eye Improves Accuracy in the Basketball Free Throw; or de Oliveira, et al, 2008, Gaze Behavior in Basketball Shooting).
Lines 103-103 are probably wrong.
ANSWER: corrected. (it was a Mendeley bug).
Methods
There is no information about the sample size estimation. This is mandatory information for this type of article.
ANSWER: Info added. Lines 124 – 130.
A control/placebo group is missing in the experimental design. Without this group, the absence of differences between groups (reported later in the article) cannot be suitable. What do you think about including this group?
ANSWER: Experimental design is used to identify whether an independent variable has an effect upon a chosen dependent variable (Jones 2015, Research methods for sport studies, Routledge, 3rd ed., p. 113). Therefore, one has to be able to manipulate the independent variable to identify such effect. In our case, we manipulated the independent variables using constant and variable practice conditions. This kind of manipulation has been the most common in studies on variability of practice (e.g. McCraken and Stelmach, 1977, A test of the schema theory of discrete motor learning, Journal of Motor Behavior; Shea and Kohl, 1991, Composition of Practice: Influence on the Retention of Motor Skills; Catalno & Kleiner, 1984, Distant transfer in coincident timing as a function of variability of practice." Perceptual and Motor skills; and many others, see e.g. Schmidt, Lee, Motor control and learning, 2005 for review).
No control group (with no practice) in the previous studies was used.
Given we have a constant and variable group (manipulation of an independent variable) we can differentiate effect these practice conditions have on gaze behavior. Control group, we assume with no practice would not add anything to our reasoning.
What is the criterion for choosing these specific distances for the trials?
ANSWER: 4.57 m is the free-throw distance in basketball. The free throw distance, as well as other distances used in our experiment, have been commonly used in studies on especial skill – skill that emerges in constant practice conditions. See:
Czyż, S. H., Breslin, G., Kwon, O., Mazur, M., Kobiałka, K., & Pizlo, Z. (2013). Especial skill effect across age and performance level: the nature and degree of generalization. Journal of Motor Behavior, 45(2), 139–52. Czyż, S. H., Kwon, O.-S., Marzec, J., Styrkowiec, P., & Breslin, G. (2015). Visual uncertainty influences the extent of an especial skill. Human Movement Science, 44, 143–149.
Fay, K., Breslin, G., Czyz, S. H., & Pizlo, Z. (2013). An especial skill in elite wheelchair basketball players. Human Movement Science, 32(4), 708–718.
Keetch, K. M., Lee, T. D., & Schmidt, R. a. (2008). Especial skills: specificity embedded within generality. Journal of Sport & Exercise Psychology, 30(6), 723–736.
Keetch, K. M., Schmidt, R. a, Lee, T. D., & Young, D. E. (2005). Especial skills: their emergence with massive amounts of practice. Journal of Experimental Psychology. Human Perception and Performance, 31(5), 970–978. Breslin, G., Hodges, N. J., Steenson, A., & Williams, a. M. (2012). Constant or variable practice: Recreating the especial skill effect. Acta Psychologica, 140(2), 154–157.
There is no reason we should use other distances.
Why a retention measure was not included?
ANSWER: We applied the procedure by Breslin, G.; Hodges, N.J.; Steenson, A.; Williams, A. M. Constant or variable practice: Recreating the especial skill effect. Acta Psychol. (Amst). 2012, 140, 154–157.
They used a pretest-posttest procedure. We mentioned it in line 99 - 100.
You included effect sizes in tables 2 and 3, but there is no information about it in this section.
ANSWER: We originally provided details in lines 210-212. We added details in lines 213 and 217 – 218.
Results
Please, standardize the number of digits after the comma in all tables (I suggest 3).
ANSWER: amended. 2 digits after comma, unless we report exact p-value (indicating less than 0.001.
Without the information about what effect size was calculated, it is impossible to check the effect of the intervention only based on the available numbers.
ANSWER: We provided details in lines 210-212. We added details in lines 213 and 217 – 218.
In general, the order of the tables and the results is not the best. You should point out in the methods section all comparisons made, and at the results section, you should follow the same order, presenting just one table/figure for each analysis. There is a massive number of tables, which, in my opinion, reduces the comprehension of the article.
ANSWER: sorted ad the we clarified the secondary objective.
If the investigation of the correlation between total fixation duration and shot efficiency is an aim of the study, it should be presented much earlier (in the introduction). Considering the current version, no rationale supports this analysis (and no hypothesis is established).
ANSWER: we rephrased the objective of the study in order to be more consistent. The secondary objective is formulated the following: “The secondary objective was to develop especial skill in constant practice conditions.” (lines 110-111) and in the discussion, accordingly.
Discussion
I strongly disagree that “practice in both conditions helped to acquire an “expert”-like gaze”. In my opinion, you don’t have enough data to support this affirmation.
ANSWER: Indeed, we agree. One study, with its limitations, is too little. We deleted this statement.
Since the choice to make a blocked assessment trial was methodologically justified, you should not use it as a possible explanation for the results (lines 348-352). If this implies problems for you, you should’ve done it differently since the beginning. So, your methodological choices are playing against your results.
ANSWER: there are only two ways testing could have been done: in random or in blocked order. If you performed the test in blocked order and then in a random one, you would change the practice load and the manipulation (the amount of practice received) would not be justified. Analogically, if done vice versa, i.e. blocked and random. The only way to avoid such a situation would be to have two more groups: constant and variable and test them in the random order. In our opinion, the methodology is correct. One can conduct an experiment using a different testing procedure. The problem is addressed in similar studies as well. (e.g. Breslin et al. 2012). Although we report the limitation, the testing procedure was not the focus of our study.
Conclusion
Concise, what is desirable in my opinion. However, I am not sure that, without a placebo group, this conclusion is, in fact, withdrawable from the results.
ANSWER: please see our answers above.
References
The issue is missing in many references.
ANSWER: corrected.
Round 2
Reviewer 2 Report
Nothing to address.